# Trends in Melanoma Phase 3 Clinical Trials since 2010: Is there Hope for Advanced Melanoma Therapies beyond Approved Treatment Mechanisms?

**DOI:** 10.3390/cancers14215184

**Published:** 2022-10-22

**Authors:** Hanna H. Kakish, Fasih Ali Ahmed, Mohamedraed Elshami, Alexander W. Loftus, Richard S. Hoehn, John B. Ammori, Lee M. Ocuin, Jordan M. Winter, Jeremy S. Bordeaux, Ankit Mangla, Luke D. Rothermel

**Affiliations:** 1Department of Surgery, Division of Surgical Oncology, University Hospitals Cleveland Medical Center, Cleveland, OH 44106, USA; 2Department of Dermatology, University Hospitals Cleveland Medical Center, Cleveland, OH 44106, USA; 3Division of Hematology and Oncology, University Hospitals Seidman Cancer Center, Cleveland, OH 44106, USA

**Keywords:** cutaneous melanoma, clinical trials, immunotherapy, targeted therapy, chemotherapy, metabolic therapy, emerging therapy

## Abstract

**Simple Summary:**

Over the last decade, the treatment of patients diagnosed with melanoma has transformed with the introduction of checkpoint inhibitors and targeted therapies. However, innate or acquired resistance, as well as toxicities of these treatments demands additional options for patients with advanced melanoma. We reviewed the landscape of phase 3 clinical trials to identify trends in clinical trial investment, as well as emerging treatments with exploratory mechanisms (mechanisms not previously approved by the FDA for melanoma) that may change the way melanoma is treated in the future.

**Abstract:**

Background: Several drugs and treatment modalities are under investigation to improve current melanoma therapy options. This review profiles the trends in clinical trial investment in late-stage melanoma, and anticipates what changes are expected in melanoma treatment, with a focus on exploratory drug mechanisms. Methods: We reviewed nine international clinical trial databases for registered, interventional, and phase 3 cutaneous melanoma clinical trials since 2010. Results: 73 trials studied drug therapies in late-stage (stage III and IV) melanoma. Exploratory mechanisms were investigated in 32% (23/73) of the late-stage melanoma drug therapy trials. Most exploratory drug trials include immunotherapy drug mechanisms (15/23 trials). Two exploratory mechanisms showed promise: the anti-LAG3 antibody, relatlimab, and the hapten modified vaccine, MVax. Many (52%) trials of exploratory mechanisms are ongoing including the use of adoptive cell transfer immunotherapies, dendritic cell vaccine therapy, and histone deacetylase (HDAC) inhibitors, among others. Conclusions: Since most clinical trials focus on previously approved drug mechanisms, it is likely that paradigm-changing treatments will involve these therapies being used in new treatment contexts or combinations. Only 2 exploratory drug mechanisms studied since 2010 have achieved promising results in the phase 3 setting, though many other trials are ongoing at this time.

## 1. Introduction

The management of cutaneous melanoma was revolutionized with the introduction of contemporary checkpoint inhibitors and MAPK targeted therapies in 2011. These treatments have replaced chemotherapy and high-dose IL-2 as standard treatment options. In 2011, the anti-CTLA4 checkpoint inhibitor antibody, ipilimumab, was approved by the U.S. Food and Drug Administration (FDA) as the first drug in history to provide a significant increase in survival for patients with advanced melanoma in a randomized phase III clinical trial [1]. Around that same time, the BRIM3 phase 3 study [2] compared the BRAF inhibitor, vemurafenib, versus the alkylating chemotherapy, dacarbazine, in previously untreated metastatic melanoma, showing improved overall survival (OS) with vemurafenib (13.2 months versus 9.6 months) [3]. In 2014, data from the CheckMate 037 trial [4] showed higher objective response rates with the anti-PD1 antibody, nivolumab, compared to chemotherapy (32% vs. 10%) after prior anti-CTLA4 failure, and led to the accelerated FDA approval of anti PD-1 therapies. Robert et al., confirmed these results using nivolumab as a first-line therapy [5]. In the 2015 OPTiM trial, the oncolytic virus, talimogene laheparepvec (T-VEC) proved to have therapeutic benefits against unresectable stage III-IV melanoma, making this the first oncolytic virus approved by the FDA [6,7]. The accelerated approval of the anti-PD1 antibody, pembrolizumab, was based on the prolonged survival of treated patients in the phase Ib KEYNOTE-001 trial [8]. The Checkmate 067 trial led to the approval of combined nivolumab and ipilimumab which significantly prolonged progression-free survival compared to single agent ipilimumab therapy [9]. Additionally, two phase 3 trials (COMBI-d and COMBI-v) demonstrated improved effects of the combination of BRAF (dabrafenib) and MEK (trametinib) inhibitors compared to monotherapy BRAF inhibition for advanced melanoma [10,11]. Beyond these studies in advanced unresectable melanomas, the benefits of these drugs have been proven in the adjuvant setting [12,13,14,15,16], dramatically altering the management of patients with resected disease at high-risk for recurrence (Figure 1).

The prognosis for many patients with late-stage (stage III and IV) melanoma has changed dramatically during this time. A result of the success of these checkpoint inhibitor immunotherapies and targeted therapies is an overshadowing other clinical trial efforts to improve outcomes for patients with melanoma. Nonetheless, objective responses from single-agent PD-1 checkpoint inhibitors occur in only 40% of patients [16]. The response rate increases to 60% with combination (anti-CTLA4/anti-PD1) checkpoint inhibitor therapy, but comes with significant toxicities in 60% of treated patients [17]. Moreover, 43% of responders ultimately develop resistance [18]. BRAF and MEK targeted therapies are an option for the 40–50% of melanoma patients with BRAF mutated tumors, however drug resistance is a predictable event within the first year of treatment [19]. As such, new treatment options including alternative mechanisms of action are needed to fill this gap in therapy.

The goal of this study was to review trends in clinical trial investments with a focus on emerging treatments that may change the way melanoma is treated in the future. We utilized ClinicalTrials.gov and eight other websites from international trial registries to find phase 3 melanoma trials that have occurred since the introduction of contemporary immune and targeted therapies. This review of phase 3 clinical trials performed since 2010 reveals the most promising emerging therapies, and provides insights into drug mechanisms and research directions that could further revolutionize melanoma care.

## 2. Materials and Methods

All phase 3 trials were collected at a single time in 11 April 2022 from 9 clinical trial databases. These include: Clinicaltrials.gov, European Clinical Trials Register, Australian New Zealand Clinical Trials registry, Registro Brasileiro de Ensaios Clinicos (the Brazilian registry of clinical trials/ReBEC), Deutsches Register Klinischer Studien (German Clinical Trials Register), Clinical Trial Registry-India (CTRI), Iranian Registry of Clinical Trials, Chinese Clinical Trial Registry (ChiCTR), and the Japanese University Hospitals Medical Information Network (UMIN)-Clinical Trials Registry (Details of these last group of trials were abstracted from the International Clinical Trials Registry Platform (ICTRP) due to difficulty with navigation of the parent website). Available data on the website was collected and organized into an excel spreadsheet (Appendix A).

Inclusion and exclusion criteria: We searched the clinical trial registers for the following criteria: cutaneous melanoma, interventional, phase 3 or 2/3, any status (completed, suspended, terminated, withdrawn, not authorized, unknown status, ongoing, or any similar phrases); stopped trials with start date on or after 1 January 2010 were included (the date used was the date of first participant enrollment, when found). We started by searching ClinicalTrials.gov, then each of the other websites using similar inclusion criteria. Trials in English, or those that could be translated to English, were included. We did not exclude expanded access trials, and trials including multiple tumor types of which cutaneous melanoma was a part. We removed duplicate trials (reported on two separate trial registries), and trials with exclusively mucosal or uveal melanoma. Finally, trials treating only adverse events of drug therapies and other supportive trials (not relating to use of a specific treatment modality) were excluded (Figure 2).

Variables: Trials were explored and classified by the following information: NCT identifier (or other trial identifier), public title, official title, start date year, trial status, reason for stopping a trial, finding agency and available results. Treatment modalities of the experimental arms were categorized into surgery, radiation, or drug therapies. Sub-classification of drug therapies occurred by the stage of melanoma being investigated (“early-stage” versus “late-stage”), then the mode of drug delivery (systemic vs. locoregional), and ultimately whether the drug mechanisms were “approved”, “versus”, “exploratory”.

Funding agency describes the organizations that provide funding or support for a clinical study. These include organizations listed as sponsors and collaborators for a study. These were classified into industry, U.S. National Institutes of Health (NIH), U.S. federal agencies (for example, Food and Drug Administration, Centers for Disease Control and Prevention, or U.S. Department of Veterans Affairs), and all others (including individuals, universities, and community based organizations).

Stage: “Late-stage” included any trial studying stage III and/or IV melanoma, and “early-stage” trials involved stage I and II only. Mode of drug delivery: Locoregional therapies are defined as drugs affecting the tumor at its primary site or regionally. Examples included intralesional, intranodal, limb infusion, and topical therapies. Drug mechanisms: Approved mechanisms included FDA approved options such as the checkpoint inhibitor immunotherapies (anti-cytotoxic T lymphocyte associated protein-4 (CTLA-4), anti-programmed cell death-1 (PD-1), and anti- programmed cell death-ligand 1 (PD-L1), BRAF/MEK targeted therapies (referred to as MAPK signaling pathway inhibitors), and the intratumoral talimogene laherparepvec (T-VEC) therapy. This group also included previously approved treatments such as high dose IL-2, interferon α, and alkylating chemotherapies. Exploratory drug mechanisms were those that were not yet approved by the FDA prior to 2022, and are only being given to patients as part of clinical trials. Further breakdown of these mechanisms characterized the drugs into “immunotherapy”, “targeted therapy”, “chemotherapy”, “metabolic therapy”, and “other” (mechanisms that did not fit the previous groups listed).

Trials were additionally classified by the treatment context (neoadjuvant, adjuvant, or metastatic/unresectable), and monotherapies versus combination therapies, when appropriate. Overlap between trials at certain points of classification were handled by including trials in all related groupings (thus, a single trial can be included under two different categories).

Interpretation of trial outcomes: For a drug to be considered “beneficial” in a phase 3 trial the intervention was required to show a better effect than the control group [20]. We reported the results of our review based on the efficacy outcomes of phase 3 trials studying exploratory drug mechanisms. Categorization and trial interpretation was performed and approved by at least two reviewers (H.K. and L.R.) with inconsistencies or deliberations clarified by a third reviewer (A.M.).

## 3. Results

A total of 288 trials were identified by initial search terms from the 9 clinical trial databases. Trials were excluded if they were duplicate trials (86), non-melanoma cancers or other diseases (69) started prior to 2010 if stopped at the time of trial abstraction (16), uveal (10) or mucosal (1) melanoma only, other reasons (7). We included 99 trials, of which 93% of these trials were listed on clinicaltrials.gov at initial search. These trials included 73 late-stage drug therapy trials (74%), 13 early-stage drug therapy trials (13%), 8 surgery trials (8%), and 6 radiotherapy trials (6%), with one trial studying the combination of radiotherapy and a drug therapy.

### 3.1. Late-Stage Trials (Stage III and IV)

Seventy-three drug therapy trials involved patients with late-stage disease (73/86; 85%), and these were described by the number of trials per drug mechanism category per year during the study period in Figure 3, and per mechanisms studied in Figure 4. Sixty-four trials (64/73; 88%) included systemic therapies and 12 trials (16%) studied locoregional therapies, with 3 overlapping trials between the two groups. Targeted therapies comprised many of these advanced trials in the early part of the study timeline, testing multiple generations of BRAF and MEK inhibitors in various treatment contexts. Immunotherapy trials have ramped up since 2010 and constitute the bulk of trials initiated and ongoing during the study period. In this same time frame, trials utilizing chemotherapy became less common internationally. Throughout the period of study, only six phase 3 trials involved different therapeutic mechanisms than immune, targeted, and chemotherapies. Three of those trials used metabolic modulators (IDO-1 inhibitors, and HDAC inhibitors), while three trials used “other” mechanisms [21,22,23] including vitamin D as a cell cycle inhibitor [24], beta blockers to limit catecholamine exposure for the tumor [25], and PV-10 as an intralesional therapy to disrupt cell lysosomes [26] (Appendix A).

#### 3.1.1. Approved vs. Exploratory Therapies

Approved therapy trials comprised 75% (48/64) and 42% (5/12) of systemic and locoregional trials, respectively. Approved mechanisms were studied in 72% (38/53) of immunotherapy trials, 91% (21/23) of targeted therapy trials, and 100% (6/6) of chemotherapy trials.

##### Locoregional: Approved vs. Exploratory

Five locoregional treatment trials used previously approved mechanisms: 3 trials using TVEC, 1 trial with interferon β and 1 intra-arterial therapy with an anti-PD1 antibody (+/− anti CTLA-4). Exploratory locoregional therapies (7 trials) include: 6 immunotherapies including cytokines (Toll-like receptor-9 (TLR-9) agonists), vaccine therapy (dendritic cells), T-cell immune stimulant drug (Bacillus Calmette Guerin (BCG) and daromun (L19IL2 + L19TNF)), in addition to one “other” mechanism (the intratumoral lysosomal disruptor [26], PV-10), (Appendix A).

##### Systemic: Approved vs. Exploratory

Approved mechanisms were used in 48 systemic therapy trials, and these involved PD-1/PD-L1 inhibitors in 52%, CTLA-4 inhibitors in 33%, MAPK pathway inhibitors in 44%, IL-2 and IFN cytokine therapies in 8%, and alkylating chemotherapies in 8%. 58% trials tested combination therapies. Multidrug combination therapies using PD-1 (or PD-L1)/BRAF/MEK inhibitors were used in two trials, showing improvement in progression free survival compared to dual combination therapies [27,28]. 50% of all approved systemic therapy trials are ongoing at the time of this review.

Exploratory systemic therapies were used in 16 trials. Nine trials (56%) involved exploratory immunotherapies such as the immune checkpoint inhibitor (ICI) anti-Lymphocyte Activation Gene-3 (LAG-3) [29,30], a T-cell immune stimulatory drug (BCG) [31,32,33], vaccine therapies (MVax [31,32], POL-103A [34], dendritic cells [35,36], Melanoma vaccine modified to express HLA A2/4-1BB ligand [37], IO102-IO103 [38]), adoptive cell transfer (ACT) immunotherapy [35,39], and cytokine therapy (granulocyte-macrophage colony-stimulating factor (GM-CSF [35,40])). Two trials used exploratory targeted therapy trials (1 multiple tyrosine kinase (MTK) inhibitor—Masitinib [41], and 1 vascular endothelial growth factor receptor inhibitor [42] (VEGFR)). Three trials used exploratory metabolic therapies including indoleamine 2, 3-dioxygenase 1 (IDO-1) inhibitors [34,43], and histone deacetylase (HDAC) inhibitors [44]. Finally, 2 trials with “other” therapy mechanisms were investigated (cell cycle inhibitor (vitamin D) and beta blockers [21,22,23]). Notably, these trials studying “other” systemic mechanisms applied these drugs as adjuvant therapies in the prevention of recurrence after complete resection.

##### Funding: Approved vs. Exploratory Trials

Approved mechanism trials were funded by industry 78% (39/50 trials) of the time with 30% (15/50) of trials involving other funding sources. (4 trials received funding from both sources). Exploratory mechanism trials received industry funding for 70% (16/23) of trials, with other funding applied to 35% (8/23) of trials. (1 trial received funding from both sources).

#### 3.1.2. Trials Investigating Exploratory Drug Mechanisms with Positive Results

Of the 73 drug trials in late-stage disease included in this review for cutaneous melanomas, 23 trials include exploratory mechanisms and 52% of these trials are still ongoing. Results were available for review for 7 of 23 trials using exploratory mechanisms. Two showed a benefit (NCT00477906, NCT03470922), and 5 showed no beneficial outcome (NCT02752074, NCT01546571, NCT02288897, NCT03445533, NCT03329846). The mechanisms showing benefit include the anti-LAG-3 checkpoint inhibitor, relatlimab, and the hapten modified MVax vaccine (mixed with BCG and used with cyclophosphamide and IL-2). On the other hand, an IDO-1 inhibitor, GM-CSF (combination and monotherapy), intratumoral TLR-9 agonist combined with systemic anti-CTLA4, PV-10, and POL-103A vaccine have not shown benefits in their respective trials.

#### 3.1.3. Treatment Context (Unresectable/Metastatic; Adjuvant; Neoadjuvant)

In regard to context of treatment; trials were divided into metastatic/unresectable, adjuvant, and neoadjuvant disease trials. The majority of trials involved metastatic or unresectable disease (52 trials; 71%). The rest studied drug in the adjuvant (18) or neoadjuvant (3) contexts.

For the 18 trials studying adjuvant therapies for late-stage disease, approved mechanisms were employed in 12 of these trials: 9 immunotherapy mechanisms (ICI and cytokine therapies) and 3 targeted mechanisms (MAPK signaling pathway inhibitors). 4 trials used approved drug mechanisms as adjuvant combination therapies. Adjuvant trials using exploratory mechanisms (6 trials) involved 4 immunotherapies including an immune checkpoint inhibitor (1 trial; anti-LAG3), and vaccine therapies (3 trials: POL-103A, Melanoma vaccine modified to express HLA A2/4-1BB ligand, dendritic cell), as well as 2 trials using “other” therapies (beta blockers, vitamin D). One trial used these exploratory drug mechanisms as a combination therapy (anti-LAG3 + anti-PD1), (Appendix A).

Neoadjuvant approaches were used in 3 trials (4%). These involved of 2 trials using the intralesional cytokine L19IL2/L19TNF (daromun) which started in 2016 and 2018 respectively. A third trial with systemic anti-PD1 and anti-CTLA4 checkpoint inhibitor combination therapy in the neoadjuvant setting was started in 2021. At the current analysis outcome data for these trials is unavailable (Appendix A).

#### 3.1.4. Combination Therapies

Approved checkpoint inhibitor immunotherapies (ICI) were used as combination with approved or exploratory drug mechanisms in 25 trials, with Table 1 summarizing the combinations with exploratory drug mechanisms. BRAF/MEK inhibitors were used as combination in 15 trials. Approved MAPK signaling pathway inhibitors were only used in combination with approved ICI, or in cross over trials with alkylating agents. Trials using drugs with exploratory mechanisms of action include combinations with approved therapies in 12 of 21 trials (57%). All of these trials involved combinations with either ICI or IL-2 immunotherapies as the approved treatment.

## 4. Discussion

The paradigm for treatment of late-stage melanoma has changed since the approval of checkpoint inhibitor immunotherapies and BRAF targeted therapies in 2011. This review of phase 3 clinical trials for late stages of melanoma since 2010 evaluates trends in clinical trial investment to highlight areas of progress and understand opportunities for future research. The focus on exploratory mechanisms of drug therapies beyond approved drug mechanisms reveals the limited progress that has been achieved apart from standard anti-PD1, anti-CTLA4 immunotherapies, and BRAF and MEK inhibitor targeted therapies.

### 4.1. Approved Mechanisms

The majority of phase 3 drug trials studied the use of approved drug mechanisms (50/73). 19 (of 50) were funded by Bristol-Myers Squibb or Merck Sharp & Dohme LLC due to their use of Yervoy, Opdivo, or Keytruda. Approved immunotherapies (mostly anti-CTLA4 and anti-PD1 antibodies) were applied as monotherapies or served as the backbone for all combinations with exploratory agents. This abundance of trials utilizing approved checkpoint inhibitor immunotherapies suggests that near-term progress in the treatment of melanoma will likely come through the use of these therapies in different treatment contexts or in combination with adjunct therapies. Supporting this trend is the recent initiation of trials in the neoadjuvant setting, and early results showing benefit for combination therapies such as pembrolizumab and lenvatinib for refractory melanoma cases [45]. Additional trials studied the ideal sequencing of immune and targeted therapies for advanced BRAF mutated melanoma and have shown superiority in survival and response rates when ICI are used as first-line over BRAF/MEK targeted therapies [46].

### 4.2. Exploratory Mechanisms

The abundance of approved mechanism trials does not diminish the likelihood that new or different therapeutic mechanisms will improve outcomes for patients with melanoma. By reviewing phase 3 trials, we identified exploratory drug mechanisms that have successfully progressed through earlier phase trials. These exploratory mechanisms are studied as monotherapies in 9 of 21 trials, and most often in patients with metastatic or unresectable melanoma (15 of 21 trials). Immunotherapies remain the most abundant category of exploratory systemic therapies (Table 2) with mechanisms including ICI, vaccine therapies, and adoptive cell therapies amongst others. Exploratory targeted therapies included multi-tyrosine kinase inhibitors, and VEGF(R) inhibitors. No phase 3 chemotherapy trials were considered exploratory as each of these trials used alkylating agent chemotherapies. (A brief review of phase 2 trials shows 68 trials of exploratory chemotherapeutic agents, 20 of which are ongoing at the time of the writing of this manuscript). Metabolic modulators are relatively new to melanoma treatment and so far have included IDO1 inhibitors and HDAC inhibitors in phase 3 trials. “Other” mechanisms were only used in the adjuvant setting, and included the use of the beta blocker, propranolol, or vitamin D as therapies to prevent recurrence. Four trials of exploratory locoregional therapy mechanisms used immunotherapies, and 1 trial involved an intralesional therapy, PV-10, categorized as “other”.

### 4.3. Promising Therapeutics

To judge progress with exploratory mechanisms, we reviewed the 7 trials with published results. The studied outcomes varied, but were most commonly disease-specific or recurrence-/progression-free survival. We hereby reviewed these exploratory mechanisms in regard to their outcome.

### 4.4. Exploratory Drug Mechanisms Showing Positive Results

Two trials using drugs with exploratory mechanisms achieved beneficial results during this contemporary time period. First, relatlimab, an antibody blocking the inhibitory immune checkpoint LAG-3, was approved in 2022 for the treatment of metastatic or unresectable melanoma in combination with nivolumab [47]. It is notable that this is the only immune checkpoint inhibitor to receive approval since the introduction of PD-1 inhibitors. The trial compared relatlimab plus nivolumab versus nivolumab monotherapy as first-line therapy for unresectable and metastatic melanoma. Median progression free survival (primary end-point) was 10.1 months for the combination therapy compared to 4.6 months with nivolumab alone, thus demonstrating its effectiveness [30]. A second trial using relatlimab was an expanded access trial that has subsequently concluded. Although this combination showed superiority to nivolumab monotherapy, efficacy and toxicities have not been directly compared with the approved dual checkpoint inhibitor therapy (ipilimumab (anti CTLA-4 antibody)/nivolumab) at this time and longer follow up time is needed to judge the effects on overall survival (secondary end-point).

Another trial with positive results in stage IV melanoma patients used a hapten modified vaccine, MVax (a vaccine prepared form the patients’ own cancer cells mixed with BCG and used in combination with cyclophosphamide followed by IL-2) [31]. The comparator arm of this trial used a similar combination of adjunct medications but without the vaccine. Primary endpoints were best overall anti-tumor response and overall survival at 2 years. Systemic use of MVax resulted in 11 anti-tumor responses (out of 97 patients) with 6 of those occurring in patients with lung metastasis (2 complete and 4 partial responses). Although the percentage of patients responding was small, these patients went on to show 21.4 months median overall survival compared to the control group with 8.7 months.

### 4.5. Exploratory Drug Mechanisms Showing Negative Results

Five exploratory drug mechanism trials have not achieved their intended results. For instance, a phase 3 trial using the POL-103A vaccine for melanoma patients after complete resection of their disease showed no benefit to RFS for its cohort of stage IIB-IIIC patients (Hazard ratio (HR) = 0.881 (95% CI: 0.629 to 1.233)) [34]. Efficacy was touted, however, in a subgroup analysis of stage IIB-IIC melanoma patients younger than sixty years old (Hazard ratio (HR) = 0.32 (95% CI: 0.12 to 0.86)) as well as those with an ulcerated primary melanoma (HR = 0.49 (95% CI: 0.26 to 0.95)). OS analysis had low sensitivity, yet the authors report HR favorable to the vaccine arm (HR = 0.64 (95% CI: 0.34 to 1.18)), and just like the RFS, the OS effects was stronger in stage IIB/IIC patients (HR 0.338 (95% CI: 0.117 to 0.975)). Further studies are needed to prove the benefit of this therapy for this highly selected population of high risk stage II patients.

Additionally, two trials studied inhibitors of indoleamine 2,3-dioxygenase 1 (IDO1), an enzyme that catalyzes the oxidation of L-tryptophan achieving metabolic modulation in multiple immune pathways in the cancer microenvironment. For the first trial using epacadostat, the drug was used in combination with the PD-1 inhibitor, pembrolizumab, and compared to pembrolizumab monotherapy. The primary outcome measures were PFS and OS at 6 months, and the combination therapy showed no difference in these outcomes between the two groups (median PFS: 4.7 vs. 4.9 months; and median OS: not reached in either group for epacadostat plus pembrolizumab vs. placebo plus pembrolizumab, respectively. HR 1.13, 0.86–1.49) [48]. This result led to other IDO1 inhibitor trials achieving low accrual, or being terminated for other reasons. The second trial targeting IDO1 used lindrostat, an orally available IDO1 inhibitor, and primarily reviewed adverse events demonstrating the tolerability of this drug by showing similar toxicity rates for the combination of lindrostat and nivolumab versus nivolumab alone [43].

In the locoregional space, a trial studying second line use of intratumoral Toll-like receptor 9 (TLR-9) agonist combined with ICI versus approved ICI monotherapy was terminated due to the lack of overall survival benefit (NCT03445533). Additionally, use of PV-10, an intralesional drug that causes lysosomal disruption, was terminated in phase 3 due to inadequate enrollment. PV-10 was compared to previously approved systemic or local therapies (TVEC), and available results did not improve on the primary outcome of progression free survival compared to the current management options (6.1 months in the experimental PV-10 arm versus 8.6 months in the comparator arm at a median follow up of 26.6 weeks (maximum of 125.8 weeks)) (NCT02288897).

A trial using autologous dendritic cell-tumor cell immunotherapy suspended in GM-CSF was terminated, although the reason for this was not clearly reported [35; NCT01875653]. A previous phase 3 trial that started in 2000 using a peptide vaccine versus GM-CSF alone, or the combination of the two which was considered limited by an ineffective vaccine [49]. Other experience with GM-CSF included an institutional report of GM-CSF in combination with ipilimumab which did not show significant improvement in overall or recurrence free survival, but did suggest a decrease in toxicities compared to historical experience. [50] One other phase 3 trial (NCT02339571) evaluating sargramostim (a GM-CSF) in combination with ipilimumab and nivolumab versus the dual checkpoint inhibitor combination is still ongoing.

It is worth noting that when a trial does not show benefit for a certain drug, this does not render that drug useless. Rather, efforts are often employed to study these drugs in different treatment contexts (e.g first-line metastatic instead of second-line) and combinations to best define the effectiveness of these therapies.

### 4.6. Ongoing Trials

Despite the limited success of exploratory therapies, a slew of additional treatment mechanisms are being studied in ongoing trials and may pique the interest of patients and clinicians when approved therapies are not an option. Dendritic cell vaccine therapies, for example, have shown promise in phase 2 trials and continue on in phase 3 [51]. VEGFR blockers which are efficacious for other cancers [52] also show promising results in early phase trials in melanoma [53]. Other ongoing phase 3 trials involve immunotherapies such as adoptive cell transfer immunotherapies, other vaccine therapies, TLR-9 agonists in combination with anti-CTLA-4 and anti-PD-1 medications as first-line treatment, metabolic modulation with HDAC inhibitors, neoadjuvant local daromun and recurrence prevention treatments using vitamin D and propranolol. Beyond these, other drugs like nilotinib, a multi tyrosine kinase inhibitor, have shown promise in phase 2 trials but have not yet moved on to phase 3 [54].

### 4.7. Opportunities for Progress

The trajectory of melanoma research is pushing heavily (and understandably) on expanding the indications for and the effectiveness of established treatment mechanisms, namely with checkpoint inhibitor immunotherapies. Based on this trend, it is likely that new approvals for melanoma treatment options will come through these existing therapy options as opposed to the use of exploratory drug therapies. Nonetheless, approved treatment mechanisms are not tolerated by all patients, due to drug resistance or treatment-related toxicities. No standard exists beyond first-line therapies for these patients, and the clinical trials listed here represent the best hope to fill that void. This review highlights gaps in the progress of research in melanoma. For instance, preclinical data shows the potential for metabolic modulators to overcome resistance to BRAF and MEK inhibitors [55] leading to early phase clinical trials, but no phase 3 trial have been initiated [56,57,58]. Evidence also exists that certain drug combinations can render melanoma cells sensitive to systemic chemotherapies, which could expand the treatment options for these patients [59] significantly due to the ease of access to these chemotherapies in cancer care. This is of interest, especially in the second-line for metastatic or unresectable disease when patients fail first-line therapies. Beyond these, new insights are evolving in the use of other oncolytic virotherapies [60], innate immunotherapies, gene-based targeted therapies, micro RNA therapies [61], PARP inhibitors [62], the fecal microbiome [63], and more.

### 4.8. Limitations

Review of clinical trial registries is limited to searchable results available on the websites which depends on accurate and updated data input. Ancillary searches for published materials regarding trial outcomes was performed to supplement the information found on the clinical trial registry sites, however reliability of further information remained questionable when presented in a non-peer-reviewed form. Furthermore, this review required data collection at one point in time, however trial updates are dynamic and may have changed by the time of publication of this manuscript. Furthermore, we were limited on trial updates, as some trials are listed as having an “unknown” status, and others having started since the 1990’s and are still on going on the website. Finally, phase 3 trials represent the most likely source of therapy options near approval, however earlier phase trials include a much wider range of drug mechanisms that may be more likely to reveal new or effective therapies that are emerging.

## 5. Conclusions

The trend in phase 3 drug trials in melanoma shows that the majority of trials are studying previously approved mechanisms of action, suggesting that paradigm-changing melanoma treatments will likely involve these therapies being used in new treatment contexts or combinations. Over the past 12 years, 32% (23 trials) of the late-stage drug trials included exploratory mechanisms, either in a locoregional (7 trials) or systemic (16 trials) manner. These have led to the approval of only one exploratory drug therapy (relatlimab; anti-LAG3 checkpoint inhibitor) during that time period, and only one additional trial has shown benefit. Nonetheless, multiple exploratory drug mechanisms are being studied in ongoing trials, with hope to expand treatment options in the near future.

## Figures and Tables

**Figure 1 cancers-14-05184-f001:**
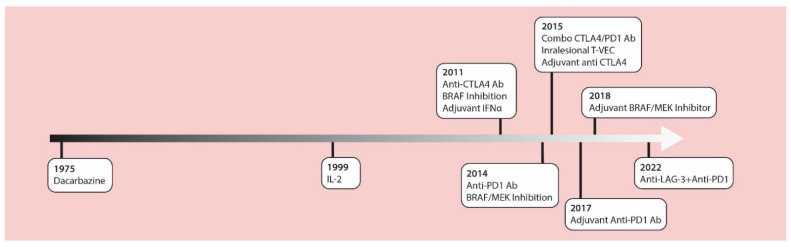
A timeline of FDA approved therapies for late-stage melanoma. IL-2 = interleukin-2; IFNα = interferon alpha; CTLA-4 Ab = cytotoxic T lymphocyte associated protein-4 antibodies; LAG-3 = lymphocyte activation gene-3; PD-1 Ab = programmed cell death-1 antibodies; T-VEC = talimogene laherparepvec.

**Figure 2 cancers-14-05184-f002:**
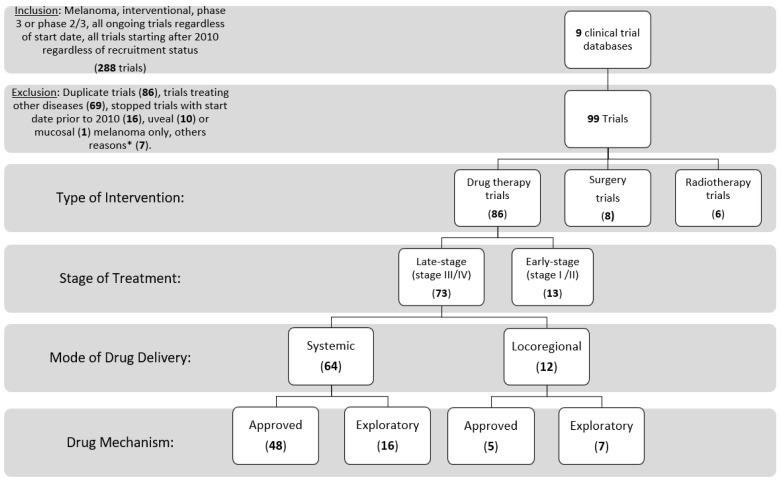
Consort diagram of trial inclusion and exclusion, and categorization. * Other reasons include having trials still listed as phase 2 on clinicaltrials.gov (3), Extension study of an included trial (1), phase 4 (1), anticipated start date 1987 (1), and a supportive trial (1).

**Figure 3 cancers-14-05184-f003:**
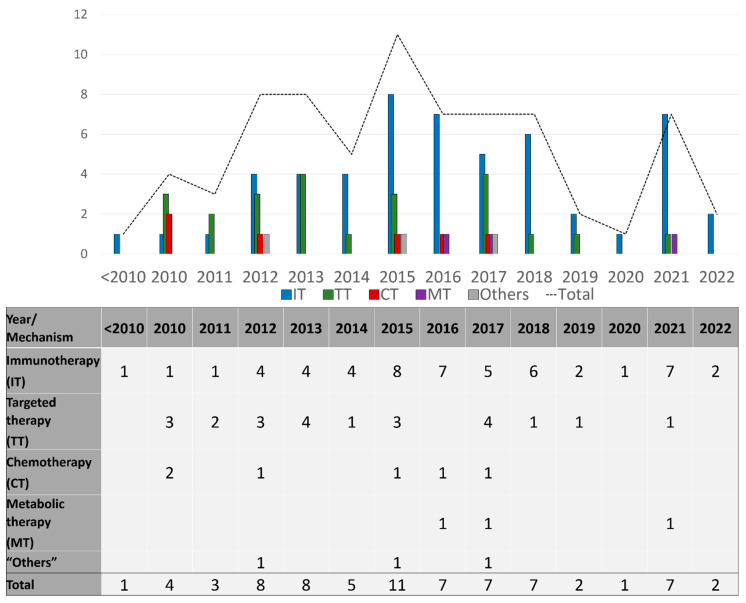
Number of late-stage phase 3 trials initiated each year, categorized by drug category.

**Figure 4 cancers-14-05184-f004:**
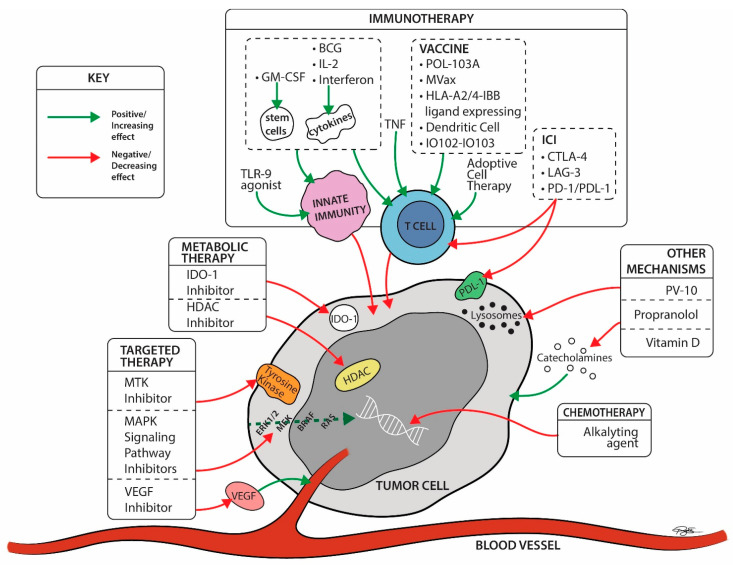
Late-stage trial drug mechanisms by site of action ©2022 University Hospitals Health System Inc. IO102 = a peptide vaccine derived from the enzyme IDO, and the montanide ISA-51. IO103 = a peptide vaccine derived from the tumor associate antigen (TAA) programmed cell death-1 ligand 1 (PD-L1). BCG = Bacillus Calmette-Guerin; CTLA-4 Ab = cytotoxic T lymphocyte associated protein-4 antibodies; ERK = extracellular signal-regulated kinase; GM-CSF = granulocyte monocyte colony stimulating factor; HDAC = histone deacetylase; IDO-1 = indoleamine 2, 3-dioxygenase 1; IL2 = Interleukin 2; LAG-3 = Lymphocyte Activation Gene-3; MAPK = mitogen-activated protein kinases; MTK = multiple kinase inhibitor; PD-1 Ab = programmed cell death-1 antibodies; PD-L1 = programmed cell death-ligand 1; RAS = rat sarcoma; TLR-9 = Toll like receptor-9; TNF = Tumor necrosis factor; VEGFR = vascular endothelial growth factor receptor.

**Table 1 cancers-14-05184-t001:** Exploratory drug therapy mechanisms in combination or monotherapy trials.

Type of Trial	Combination Mechanisms	Single Mechanisms
Late-stage Systemic	IDO-1 inhibitors + anti-PD-1 HDAC inhibitors + anti-PD-1 Anti-LAG3 + anti-PD-1 IO102-IO103 vaccine * + anti-PD-1 MVax + BCG + cyclophosphamide + IL2 VEGFR inhibitor + anti-PD-1 GM-CSF + anti-CTLA4 + anti-PD-1 Dendritic cell Vaccine + ACT **	MTK inhibitor POL-103A Vaccine Melanoma vaccine modified to express HLA A2/4-1BB ligand ACT Beta blockers Vitamin D
Late-stage Locoregional	TLR-9 agonist + anti-CTLA4 or anti-PD-1 IL-2 and + BCG	PV-10 DC vaccine Daromun (L19IL2/L19TNF)

* IO102, a peptide vaccine derived from the enzyme IDO, and the montanide ISA-51. IO103, a peptide vaccine derived from the tumor associate antigen (TAA) programmed cell death-1 ligand 1 (PD-L1) [38]. ACT = adoptive cell transfer therapy; BCG = Bacillus Calmette-Guerin; CTLA-4 Ab = cytotoxic T lymphocyte associated protein-4 antibodies; DC = dendritic cell; GM-CSF = granulocyte monocyte colony stimulating factor; HDAC = histone deacetylase; IDO-1 = indoleamine 2, 3-dioxygenase 1; IL2 = Interleukin 2; LAG-3 = Lymphocyte Activation Gene-3; MTK = multiple kinase inhibitor; PD-1 Ab = programmed cell death-1 antibodies; TLR-9 = Toll like receptor-9; VEGFR = vascular endothelial growth factor receptor. ** This trial involved exploratory mechanism combination without a previously approved therapy.

**Table 2 cancers-14-05184-t002:** Number and types of exploratory therapy trials per category classified by mode of delivery.

Exploratory Therapies	Immunotherapy	Targeted Therapy	Chemotherapy	Metabolic Therapy	Others	Total
Systemic	9	2	0	3	2	16
	Anti LAG-3 Vaccine ^a,d,e^ Cytokine ^a,b,c^ ACT Immunostimulatory T cells ^a^	MTK Anti-VEGFR		IDO1 inhibitors HDAC inhibitors	Vitamin D Beta blockers	
Locoregional	6	0	0	0	1	7
	Cytokine ^f,h,k^ Vaccine ^g^ Immunostimulatory T cells ^h^				PV-10	

^a^ MVax + BCG + cyclophosphamide + IL2; ^b,g^ dendritic cell vaccine; ^c^ granulocyte monocyte colony stimulating factor; ^d^ Melanoma vaccine modified to express HLA A2/4-1BB ligand; ^e^ IO102-IO103; ^f^ TLR-9 agonist; ^h^ IL-2 + BCG; ^k^ Daromun (L19IL2 + L19TNF).

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
