# Peer review of "Trends in Melanoma Phase 3 Clinical Trials since 2010: Is there Hope for Advanced Melanoma Therapies beyond Approved Treatment Mechanisms?"

_cancers, 2022, doi:10.3390/cancers14215184_

Round 1
Reviewer 1 Report
1. Overall a clear written manuscript, could you expand more on endpoints and 5-year prognosis rate on each mechanism of treatment so it would be clearer and gives direct view on the treatments.
2. Figure 4, almost all of the mechanisms are covered while need to be more specific on mainstream mechanisms, upstream and downstream signaling proteins should be mentioned and explained.
3. After MEK-BRAF inhibition there comes anti-PD1 and combination therapy, which is hot topics these days. Give more details via future opportunities on new treatments especially PD1 related treatments in discussion section.
Reviewer 2 Report
This review provides an overview of phase 3 clinical trials to explore trends in clinical trials and identify emerging treatments with exploratory mechanisms. The topic of this review is very relevant as a number of drugs and treatment modalities are being investigated to improve the current melanoma therapeutic options. This review describes trends in clinical trials investment in late-stage melanoma and predicts what developments in melanoma treatment can be expected, with a focus on exploratory drug mechanisms. The manuscript is comprehensive and well written.
The review searched nine international clinical trials databases for phase 3 cutaneous melanoma clinical trials since 2010 and found 73 relevant drug therapy trials. The Methods are correctly designed detailed. The strength of Results is that it clearly presents the approved and exploratory studies separately. The Discussion is coherent, and 7 published trials were also reviewed in the manuscript to highlight the promising therapeutic aspects of exploratory drugs.
Great paper in every aspect, accurate and detailed review.
